# OpenReview forum: "NeurIPS 2023 Competition: Privacy Preserving Federated Learning Document VQA"
_TMLR — Accepted by TMLR_

### Review · Reviewer_NWHm · 2025-02-25

**Summary Of Contributions:**

This work present the solutions to the previously held NeurIPS challenge on privacy-preserving federated document VQA. The challenge is split into two (distinct, but complementary) parts: minimal required utility threshold and differentially private learning. This work goes through the challenge itself, the solutions (with background) and lessons learnt.

**Audience:**

No

**Broader Impact Concerns:**

I am not sure on the impact of this work at all, I struggle to see what exactly is the contribution here (and how this contribution is split between the participants of the challenge and the authors of the manuscript). I am not in favour of accepting this work because a) I do not see how this is relevant to the list of topics for TMLR submissions and b) I struggle to see the useful contributions that belong to the authors here.

**Claims And Evidence:**

No

**Requested Changes:**

As above, there is no specific structure to the manuscript to describe what the contribution is. In my view the work itself requires a serious rewrite based on what specifically is the message the authors want to convey.

**Strengths And Weaknesses:**

This is a very unusual work - I am actually struggling to identify any area from the potential contributions list from the TMLR submission website, so in my view this work does not necessarily fall into the scope of TMLR. Moreover, it is not immediately clear to me which specific part of this work was done by the authors and by the submitted works during the challenge. The references to 'we' are somewhat misleading unless everything was done by the authors (which judging by the context is not the case?). Nonetheless, I will leave the decision on the suitability to the AE and will provide the objective/actionable feedback on the work itself.

The problem that the work studies is very relevant and important for both the fields of FL and document-based VQA. However the actual analysis of the problem, approaches and the results are in my view on a shallow side. I additionally had a hard time understanding the work itself - the structure is convoluted, there is no easy way to distinguish what the work ACTUALLY proposed. There are some tasks to solve, solutions, takeaways, but these all seem to be completed by different people (since participants provided the solutions, not the authors from my understanding, but correct me if I am wrong). So in short: I do not actually understand what specifically this manuscript is about: lessons learnt from the challenge? Or how the challenge was approached? Or new promising direction for VQA in FL? There is a little bit on all of these topics here, but none of them are addressed in an adequate level of details.

Content-wise part 2 (differential privacy) demonstrates the issues I have with the manuscript very well. Your background on DP is very limited: you jump straight into eps-delta DP without ANY explanation of the principles behind DP, pure-DP definition, or any background which is actually relevant to the definition (admittedly some of it is in the appendix, but there is no reference to the appendix in your background section at all). 'High utility models' under DP seems to be a semi-random collection of DP references, it lacks structure and relevance to the problem. I would advise the authors to rework DP section to a) have a concrete structure the reader can actually follow, b) establish concrete relevance to the problem at hand, because just referring the reader to other papers for both the relevant literature and the details on DP is not acceptable. There is no discussion on the relevance of DP-SGD, its issues in FL, any alternatives such as DP-FTRL - this has to be addressed. Your entire second part of the submission is based on DP, otherwise it is challenging to understand both the setting of the problem as well as what are the existing challenges which could not have previously been solved using standard approaches.

Moreover, when we actually look into the analysis you for some reason claim epsilon of 1.0 is 'medium'? Medium with respect to what? Is there any relevant literature which would call this epsilon level 'medium'? But more interestingly authors later claim that  'The baseline solution for track 2 utilizes DP stochastic optimization.' What exactly does this mean? Are you referring to DP-SGD? Because the context suggest this, but there is no specific mention otherwise. Appendix seems to refer to the established DP theorems, proofs and definitions, not to anything specifically tailored to the problem at hand up until the very end, when a new previously undiscussed definition of DP for FL appears out of nowhere. Lessons learnt with respect to DP are also not particularly novel or useful: 'needing to ensure that implementations are DP' is absolutely trivial. Asking for proofs? You already stated in the problem definition that this is required, why is this a lesson learnt at the end? No FL-specific issues are mentioned at all as well: subsampling, sync/async FL wrt DP etc. These would have been very interesting topics for discussion, but they are simply missing.

So in essence the DP section requires a SUBSTANTIAL rework with respect to having a clear and concise literature review at the start with all relevant DP definitions; methods for DP training of ML models (including those relevant to LMs like DP-Forward and those relevant to FL like DP-FTRL etc.), novel actionable insights on how to interpret DP section of the results etc.

Section on FL suffers from similar issues: not enough details on the solutions proposed, what are the new novel insights that can be gained etc. We already know that using LoRA can limit the communication cost and lead to better models, for instance. More subjectively I would have liked to have seen more details on the challenge itself (without having to look it up): how is the data distributed, is async FL tested at all etc. There is no clear setup-solution-message in this section either, but it is maybe severe compared to DP.

---

> ### Author Response · Authors · 2025-04-07
>
> > Your background on DP is very limited: you jump straight into eps-delta DP without ANY explanation of the principles behind DP, pure-DP definition, or any background which is actually relevant to the definition
>
> We expanded the background Section to include more explanation of the principles behind DP and mention DP.
>
> >  'High utility models' under DP seems to be a semi-random collection of DP references, it lacks structure and relevance to the problem.
>
> We improved this paragraph by adding that “DP introduces a trade-off between privacy and utility that can make it hard to use DP in some applications (Ponomareva et al., 2023).” and then motivating through the recent advancements why the competition used pre-trained data. “Because of these advancements and the popularity of the approach this competition also assumes the availability of a model checkpoint that has been pre-trained on public data.”
>
> > rework DP section to a) have a concrete structure the reader can actually follow, b) establish concrete relevance to the problem at hand, because just referring the reader to other papers for both the relevant literature and the details on DP is not acceptable.
>
> We tried to improve this (as discussed above).
>
> > There is no discussion on the relevance of DP-SGD, its issues in FL, any alternatives such as DP-FTRL - this has to be addressed.
>
> We added a paragraph on “Training ML models under DP” in Section 2 where we discuss DP-SGD. There we added: “Furthermore,DP-FTRL (Kairouz et al., 2021a) adds correlated noise which can in some cases improve the utility/privacy trade-off.”
> Note that the methods in Sections 4+5 that we discuss are actual methods from participants and we cannot report on any other more novel methods if they have not been submitted.
>
> > You for some reason claim epsilon of 1.0 is 'medium'? Medium with respect to what? Is there any relevant literature which would call this epsilon level 'medium'?
>
> We changed the classification of the privacy budgets based on Ponomareva et al. (2023) as follows: “Participants are required to train under DP at different levels from strong formal DP ($\epsilon=1$) to reasonable privacy guarantees ($\epsilon=8$) to mitigate the risk of provider information being leaked.”
>
> > 'The baseline solution for track 2 utilizes DP stochastic optimization.' What exactly does this mean? Are you referring to DP-SGD? Because the context suggest this, but there is no specific mention otherwise.
>
> We changed this to be more explicit and wrote: “The baseline solution for track 2 utilizes DP-SGD with provider-level add/remove adjacency.”
>
> >  No FL-specific issues are mentioned at all as well: subsampling, sync/async FL wrt DP etc
>
> We recognise that more complex settings are important to consider in future work and added to the limitations the following: “The observed settings do not focus on async/sync FL communication settings but assume simpler communication settings.”
>
> > methods for DP training of ML models (including those relevant to LMs like DP-Forward and those relevant to FL like DP-FTRL etc.), novel actionable insights on how to interpret DP section of the results etc.
>
> We added a paragraph on “Training ML models under DP” in Section 2 where we discuss DP-SGD, there we added the suggested citations: “For language models specific optimization methods like DP-Forward (Du et al., 2023) can be beneficial. [...] Furthermore, DP-
> FTRL (Kairouz et al., 2021a) adds correlated noise which can in some cases improve the utility/privacy trade-off.”
> Note that the methods in Sections 4+5 that we discuss are actual methods from participants and we cannot report on any other more novel methods if they have not been submitted. We agree with the reviewer that more advanced methods are interesting to consider in future work.
>
> > Section on FL suffers from similar issues: not enough details on the solutions proposed, what are the new novel insights that can be gained etc. We already know that using LoRA can limit the communication cost and lead to better models, for instance.
>
> We moved the Appendices discussing the methods and trade-offs to the main paper. This can be seen in the Sections 4 and 5 that are now significantly longer. We only left some larger Figures and two Algorithms in the Appendix as including these would extensively lengthen the paper. Note that the methods that we discuss are actual methods from participants and we cannot report on any other more novel methods if they have not been submitted.
>
> > more details on the challenge itself (without having to look it up): how is the data distributed
>
> We added references to the dataset distribution information in Table A1 and Figure A2 to Section 3.1. Note that Figure A2 has been newly added to the Appendix.

---

### Review · Reviewer_GHaR · 2025-03-17

**Summary Of Contributions:**

The paper presents an overview of the NeurIPS'23 competition on privacy-preserving federated learning (FL) for document visual question answering (VQA), highlighting the ML techniques (e.g., LoRA) and optimization strategies (FedShampoo) used by the winning and runner-up teams across the competition’s two tracks: communication-efficient FL and differentially private learning.  The authors have clearly shown the efficiency gains from each of the optimization steps employed by the winners and runners for both the tracks, which could serve as a useful reference for future research looking to optimize the trade-offs between privacy, utility, and communication in similar settings.

**Audience:**

Yes

**Broader Impact Concerns:**

It could have been better-articulated.

**Claims And Evidence:**

Yes

**Requested Changes:**

See the weaknesses.

**Strengths And Weaknesses:**

### Strength

$\bullet$ The authors have presented the efficiency gains from each of the ML approaches and algorithmic optimization, along with a detailed ablation study in Appendix.

$\bullet$ The authors have done a great job of explaining the difference between the two competition tracks (communication-efficient FL and differentially private learning),  making it clear why each one matters and what challenges they address.


$\bullet$ The breakdown of efficiency gains and optimization techniques used for high-efficiency gains without sacrificing the utility (in DP-settings), making it a useful resource for future research and further exploration of privacy-preserving methods in real-world settings.


### Weaknesses

$\bullet$ The biggest problem with this paper is that it’s essentially just a competition summary with no real research contribution. It doesn’t introduce any new methods or offer any deeper insights beyond the obvious takeaways from the competition results. The benefits of LoRA fine-tuning for efficiency and quantization for communication reduction are already well understood, so the paper doesn’t add anything meaningful, and seems more like a report than a research paper.


$\bullet$ The current version of the manuscript is unnecessarily short at nine pages, with important ablation studies pushed to the appendix instead of being included in the main text, making it harder to get a complete picture of the trade-offs involved in different techniques.


$\bullet$ The paper is difficult to follow, even for experts, due to its disorganized structure and lack of a clear narrative. Key takeaways are not well-highlighted, making it hard to quickly grasp the main contributions. Instead, information is scattered across dense sections, requiring readers to sift through lengthy explanations to extract relevant insights. A more structured presentation with clearer sectioning and emphasis on key findings would greatly improve readability.

---

> ### Author Response · Authors · 2025-04-07
>
> > The current version of the manuscript is unnecessarily short at nine pages, with important ablation studies pushed to the appendix instead of being included in the main text, making it harder to get a complete picture of the trade-offs involved in different techniques.
>
> Thanks for the suggestion! We moved the Appendices discussing the methods and trade-offs to the main paper. This can be seen in the Sections 4 and 5 that are now significantly longer. We only left some larger Figures and two Algorithms and the privacy analysis of the baseline of track 2 in the Appendix as including these would extensively lengthen the paper.
>
> > Key takeaways are not well-highlighted, making it hard to quickly grasp the main contributions.
> > A more structured presentation with clearer sectioning and emphasis on key findings would greatly improve readability.
>
> Thanks for the suggestion! We added takeaway boxes summarizing the main findings from Track 1 and 2 in Sections 4.5 and 5.5. We believe that the findings regarding future DP and FL competitions are already quite well summarized in Section 6 and that condensing them further would not yield more benefit.

---

### Review · Reviewer_71WV · 2025-03-17

**Summary Of Contributions:**

The manuscript discusses the various aspects of the Privacy Preserving Federated Learning Document VQA (PFL-DocVQA) competition of NeurIPS 2023. The goal of the competition was to develop comprehensive solutions for federated invoice processing that are provably private and communication-efficient. Essentially, the task was to extract information and reason over real invoice document images based on provided questions and answers, while considering the privacy-preserving scenario where the data is distributed across multiple nodes in a federated learning setting. Therefore, the PFL-DocVQA brought together people from many domains, including document analysis, privacy, and federated learning.

**Audience:**

Yes

**Broader Impact Concerns:**

I do not have broader impact concerns.

**Claims And Evidence:**

No

**Requested Changes:**

* Provide clear justification on why datasets and benchmarks should be publishable in TMLR
* Provide the model inversion attacks results

**Strengths And Weaknesses:**

Strengths:

* Starter kit was provided for the participants to understand the implementation details
* The problem of automated invoice processing in a federated setting is interesting


Weaknesses:

* It is simply a benchmark paper. I do not think TMLR is a suitable venue for the same.
* FL nodes do not have invoices from common providers, which limits scope significantly
* All techniques discussed are prior works.
* No model inversion attack seems to have been considered to demonstrate the leakage aspect in practice
* How do the organizers ensure that the submitted code is correct is not clear. Furthermore, there is not much attempt to address the DP analysis verification.

---

> ### Author Response · Authors · 2025-04-07
>
> > FL nodes do not have invoices from common providers, which limits scope significantly
>
> We added this as a limitation to Section 7 (in blue): “We do not consider settings where the data of a provider is split among multiple nodes, which would require more advanced composition methods under DP and methods to detect providers that are shared among nodes.”
>
> > All techniques discussed are prior works.
>
> Note that the methods in Sections 4+5 that we discuss are actual methods from participants and we cannot report on any other more novel methods if they have not been submitted.
> > No model inversion attack seems to have been considered to demonstrate the leakage aspect in practice
>
> We added this as a limitation to Section 7 (in blue): “Furthermore, we did not run any membership inference attacks, gradient inversion attacks or auditing methods to empirically assess the privacy leakage of the approaches.”
> > How do the organizers ensure that the submitted code is correct is not clear. Furthermore, there is not much attempt to address the DP analysis verification.
>
> In Section 6.1 we write “Thus, members of the organizing team have inspected the implementations of the best scoring methods but this is a manual process that does not scale to competitions with a large number of participants.” and discuss variants of enhancing this in the future.

---

### Comment · Action_Editor_Uv8y · 2025-03-21
**Clarification on Evaluation Criteria and Scope**

Dear Reviewers and Authors,

To ensure our discussions remain constructive, I would like to remind everyone that TMLR's primary evaluation criteria focuses on the correctness and relevance of the research (i.e., its value to some audience), while novelty is not a requirement for acceptance. Therefore, evaluations should not be based on the absence of novelty.

Regarding concerns about whether the paper falls within TMLR's scope, I believe it does—hence its current review status. However, discussions with the Editors-in-Chief (EiCs) are ongoing to clarify this aspect before any final acceptance decision. For now, I kindly ask that we set this argument aside and focus on evaluating the paper based on its merits.

Reviewer NWHm, regarding your concerns:
- "Moreover, it is not immediately clear to me which specific part of this work was done by the authors and by the submitted works during the challenge. The references to 'we' are somewhat misleading unless everything was done by the authors (which judging by the context is not the case?)."
- "I struggle to see the useful contributions that belong to the authors here."

for now, please set aside this ambiguity and focus on evaluating the paper based on other aspects. This specific concern will be reviewed and addressed during the decision-making process to ensure that all contributions listed in the paper are the authors' own work.

Best regards,

Your AE.

---

### Comment · Action_Editor_Uv8y · 2025-03-22
**Further Clarification on Scope and Evaluation Criteria**

Dear Reviewers and Authors,

After discussing with the EiCs, I would like to clarify the scope of submissions to TMLR. A paper is considered within scope if it presents clear claims backed by either theoretical or empirical justification.

Please assess the paper based on TMLR’s standard criteria (as I pointed out before):
- claims and supporting evidence
- the relevance and interest to TMLR’s audience.

Thank you for your continued efforts in upholding the quality of TMLR.

Best regards,

AE.

---

### Author Response · Authors · 2025-04-07
**Revision based on reviewers' input**

Thanks for your comprehensive feedback and suggestions. We have replied to individual points below each review but would like to highlight the major changes in the uploaded revision here (we highlighted changes to the initial version in blue in the revision):
- Expansion of Section 2 (Background): DP and DP ML Sections (NWHm)
- More details on the dataset in Section 3.1(added missing references) and new Figure A2 in Appendix (NWHm)
- Moved the details from the Appendix to Sections 4 and 5 (GHaR)
- Added Takeaway boxes regarding the submitted methods in Sections 4.5 and 5.5 (GHaR)
- Added limitations in Section 7 (71WV, NWHm)

---

### Decision · Action_Editor_Uv8y · 2025-05-12

**Recommendation:** Accept with minor revision

**Comment:**

All reviewers raised issues on the authorship as the authors wrote about winners and participants' solutions and ablations as part of the contribution of the paper. During discussion between AE and authors, authors provided information that participants and winners are actually co-authors of the paper, but this cannot be revealed due to anonymity. However, I ask authors to make a clear contributions list below in the "Requested changes" to make sure authorship is properly provided.

All reviewers agreed that the updated revision is more solid as a research paper and now articulates the message and story better. However, all of the concerns of Reviewer 71WV were articulated only as limitations, which may not be the best course of action. Part of the concerns I think is okay to keep as limitations, but some could be improved for the final revision (see "Requested changes").

The last critical issue raised by Reviewer NWHm is the helpful / interesting takeaways from the paper as the must thing for TMLR papers. While the Review agreed that readability is improved, despite the authorship issue, the Reviewer still doesn't see interesting / new / helpful takeaways. As I stated in the "Audience" section, takeaways are helpful from an empirical perspective from my point of view even having all the things which are known. However, to improve this message even more, I ask authors to expand a bit on the takeaways to make connections and refer to whether observations are in agreement or not with prior results.

**Requested changes:**
- As the paper is co-authored by both organizers and participants at the same time, and also some organizers were participants (correct if I am wrong, but also this is very biased to me as organizers had access to the closed set, thus maybe this should be prohibited by rules that participants can be from organizers' list), authors must de-anonymize the paper including all participants' names and provide their exact contributions / methods they used. It is probably good to specify in the author list exactly who are organizers, who are participants co-authoring the paper, and then provide detailed contributions in the end of the paper.
- Expand more details regarding questions raised by Reviewer 71WV. Particularly
  - "No model inversion attack seems to have been considered to demonstrate the leakage aspect in practice" -> could authors provide some references to justify that this is the real issue and leakage happens in practice (kind of obvious thing to me, but good to support with prior work reported this issue)
  - "How do the organizers ensure that the submitted code is correct is not clear. Furthermore, there is not much attempt to address the DP analysis verification." -> could authors put more details into the text how exactly organizers were double checking code and proofs (maybe running code, maybe doing code review, maybe re-implementing, etc.) - this is a key thing of the paper to share practices + raise discussion on how we could check things in the critical domains where error cost is huge and harmful.
- Expand key takeaways to make connections to currently observed empirical results (to fully address concern from Reviewer NWHm): showcase what is aligned and what is not, what is something new (if any) which is not probed before (e.g. some hyperparameters are very crucial to make things work in practice despite theory).

**Note: If changes will not be integrated I would like to retract "acceptance with minor revision" and change status to "rejection with major revision" as authorship and contributions from participants must be properly reflected.**

**Audience:**

While paper discussed the competition, which may not be interested to the TMLR community, the part related to the competition evaluation itself maybe useful to community to understand what improvements we may need to make to have the standard for the evaluation of FL and DP methods before we deploy them as they touch sensitive information. Moreover, having many papers on DP and FL from more theoretical perspective and very limited empirical evaluation, the paper can bring the gap here showcasing how all known standard things works for more realistic scenario showcasing the gap between research and deployment.

Having own experience in empirical part of FL and DP and knowing how many things are actually hard to train for more realistic tasks and even apply to standard benchmarks we have, I learned / confirmed myself some aspects so thus I believe it is useful takeaways and summary of the ablations which may navigate and inspire deeper empirical investigation in FL and DP community.

**Claims And Evidence:**

The paper is co-authored (per discussion with authors) by organizers and participants / winners of the NeurIPS'23 competition on privacy-preserving federated learning (FL) for document visual question answering (VQA) and presents overview of the competition organization including data, metrics, tracks (communication-efficient FL and differential privacy), baselines, evaluation of the final solutions and correctness checking, and participants / winners solutions and ablations.

The paper is more methodological as presents how to organize the competition and evaluate usefulness of the results (the later is really crucial as kaggle-style competitions were not super useful for real case problems in the end). E.g. organizers consider how to evaluate and ensure that privacy is guaranteed assuming that solutions are targeting real deployment for sensitive information.

Another aspect of the work is participants and winners solutions and ablations which are discussed in the paper. While all things presented and used are prior works techniques and known in the community methods like (LoRA, FedShampoo, quantization, DP variants), the paper particularly discusses how all they play for document visual question answering with federated learning and differential privacy, moreover how all of them are applicable for specific real-case benchmark which was prior proposed by the same authors.

As stated by Reviewer GHaR "The authors have clearly shown the efficiency gains from each of the optimization steps employed by the winners and runners for both the tracks, which could serve as a useful reference for future research looking to optimize the trade-offs between privacy, utility, and communication in similar settings."

---

> ### Author Response · Authors · 2025-05-23
>
> Dear AE and reviewers,
>
> we are currently preparing the revision that incorporates the requested changes and will share the updated version with you soon.
>
> Best,
> the authors

---

> > ### Author Response · Authors · 2025-06-02
> >
> > Dear AE and reviewers,
> >
> > thanks for the feedback on our revision. We will post a new revision to address the requested changes in a minute:
> >
> > > As the paper is co-authored by both organizers and participants at the same time, and also some organizers were participants (correct if I am wrong, but also this is very biased to me as organizers had access to the closed set, thus maybe this should be prohibited by rules that participants can be from organizers' list), authors must de-anonymize the paper including all participants' names and provide their exact contributions / methods they used. It is probably good to specify in the author list exactly who are organizers, who are participants co-authoring the paper, and then provide detailed contributions in the end of the paper.
> >
> > We added the authors and author contributions (and added the participants names to the sections and tables where applicable). We believe that there was a misunderstanding regarding the organizers being participants. We clarified that in the author contributions: "The participants do not have any connection to the organizers, and did not have access to the test data before the end of the competition. The participants were invited by the organizers to contribute to the writing of the manuscript."
> >
> > > Expand more details regarding questions raised by Reviewer 71WV.
> >
> > > Particularly "No model inversion attack seems to have been considered to demonstrate the leakage aspect in practice" -> could authors provide some references to justify that this is the real issue and leakage happens in practice (kind of obvious thing to me, but good to support with prior work reported this issue)
> >
> > We added references to (Tito et al., 2024; Pinto et al., 2024) to the limitations and to the formulation of Track 2 (Sec. 5.1). Both these works showed the leakage in practice.
> >
> > > "How do the organizers ensure that the submitted code is correct is not clear. Furthermore, there is not much attempt to address the DP analysis verification." -> could authors put more details into the text how exactly organizers were double checking code and proofs (maybe running code, maybe doing code review, maybe re-implementing, etc.) - this is a key thing of the paper to share practices + raise discussion on how we could check things in the critical domains where error cost is huge and harmful.
> >
> > We added a paragraph to Sec. 5.1 that discusses how the organizers reviewed the submissions.
> >
> > > Expand key takeaways to make connections to currently observed empirical results (to fully address concern from Reviewer NWHm): showcase what is aligned and what is not, what is something new (if any) which is not probed before (e.g. some hyperparameters are very crucial to make things work in practice despite theory).
> >
> > We added discussion on the in Sec. 4.5 and 5.5 linking the key takeaways to the common literature.
> >
> > Besides that we have updated the references, fixed some small typos, added links to code repositories, and added acknowledgements. We also removed the blue font that denoted the difference between initial version and first revision.
> >
> > Best, the authors

---

> > > ### Comment · Action_Editor_Uv8y · 2025-06-03
> > > **Reply**
> > >
> > > Dear Authors,
> > >
> > > Thanks for including all pieces I requested. I did a pass over the revision. Looks good to me and very comprehensive!
> > >
> > > AE.